# Effects of Dietary Glycinin on Oxidative Damage, Apoptosis and Tight Junction in the Intestine of Juvenile Hybrid Yellow Catfish, *Pelteobagrus fulvidraco* ♀ × *Pelteobaggrus vachelli* ♂

**DOI:** 10.3390/ijms231911198

**Published:** 2022-09-23

**Authors:** Linyuan Yi, Jingwen Liu, Huijun Yang, Aijie Mo, Yuxiang Zhai, Siru Wang, Yongchao Yuan

**Affiliations:** 1Key Lab of Freshwater Animal Breeding, Ministry of Agriculture, College of Fisheries, Huazhong Agricultural University, Wuhan 430070, China; 2Shuangshui Shuanglu Institute, Huazhong Agricultural University, Wuhan 430070, China; 3National Demonstration Center for Experimental Aquaculture Education, Huazhong Agricultural University, Wuhan 430070, China

**Keywords:** anti-nutritional factors, intestinal damage, antioxidant capacity, growth performance, hybrid yellow catfish

## Abstract

The objective of this study was to examine the influences of glycinin for growth and intestinal structural integrity related to oxidative damage, apoptosis and tight junction of juvenile hybrid yellow catfish (*Pelteobagrus fulvidraco* ♀ × *Pelteobaggrus vachelli* ♂). Fish (initial weight, 1.02 ± 0.01 g) were fed diets containing five different levels of glycinin at 0%, 2%, 4%, 6%, and 8% for 8 weeks. The results demonstrated that dietary glycinin levels had a negative correlation with final weight, feed intake, protein efficiency ratio and survival rate of the experiment fish. When the level of dietary glycinin exceeded 4%, the structural integrity of the posterior intestine was observably impaired, characterized by disordered and exfoliated margin of intestinal villi, blurred and broken boundaries of tight junctions, damaged organelles and cell vacuolation. Levels of 4–8% dietary glycinin depressed the total antioxidant capacity and total superoxide dismutase activities of posterior intestine. Furthermore, a high level of dietary glycinin linearly and quadratically down-regulated the mRNA expressions of Claudin-1, Occludin and ZO-1, while it linearly and significantly up-regulated the mRNA expressions of Bax, Cyt C, Caspase 3, Caspase 9 and p53 in the posterior intestine. In conclusion, dietary 4–8% glycinin impaired the morphological structure of the posterior intestine by inducing oxidative stress and cell apoptosis, and eventually impeded the growth performance of juvenile hybrid yellow catfish.

## 1. Introduction

Soybean proteins are considered as one of the superior plant proteins to substitute fish meal in aquatic compound feed because of their balanced nutrition composition and high bioavailability [1]. However, previous studies have shown that excessive soybean protein in diets would reduce the feed palatability and feed conversion efficiency of cultured fish [1,2,3,4,5,6]. These studies demonstrated that excessive soybean meal induced intestinal injury in amberjack [1], grass carp [4], abalone [6] and orange-spotted grouper [7], and caused intestinal oxidative stress and immunosuppression, eventually having a negative impact on growth and health. Currently, anti-nutritional factors (ANFs) including soybean antigen protein are partially regarded as the reasons why soybean protein inhibits growth and limits the high use of soybean meal in valuable farmed fish [8,9]. It has been confirmed that glycinin, with its strong immunogenicity, is one of the main soybean antigenic proteins, composing around 40% of the total soybean seed proteins [10]. As a thermostable sensitizing protein composed of six monomeric subunits, it has a tight structure and is difficult to be hydrolyzed by enzymes [11]. Glycinin might produce anti-nutritional effects such as damaged intestinal structure and immune system. Earlier studies have reported that soybean antigen protein could cause endothelial cell proliferation and villus damage in the small intestine of piglets and calves [12,13]. Moreover, glycinin and β-coglycinin induced the increase in IgE and IgG1 antibodies in the intestine of murine, accompanied by high histamine release and severe degranulation of mast cells [14]. The intestinal tract is fragile and vulnerable to attack from external antigens, while it directly contacts with glycinin. The influence of glycinin for the growth and intestinal health of aquatic animals has been well documented. Exaggerated additions of glycinin in the diet can induce allergic reactions and enteritis, which causes intestinal damage and the attenuation of digestive function, immune function and antioxidant capacity [15,16,17,18,19,20,21]. However, different fish species have different tolerances to glycinin. In order to adjust feeding strategies, more studies are needed on soybean meal or ANFs that consider fish species and growth stages.

It is worth noting that the integrity of intestinal construction is a necessity for the development of fish. Disruption of the intestinal structure can include compromised cell disturbance via mechanisms such as oxidative stress and apoptosis, as well as the damage of tight junction proteins, which is an important structure connecting intestinal epithelial cells [22,23]. Glycinin exposure in the diet could lead to the destruction of the intestinal physical barrier and function, characterized by the decrease in absorption vesicles, the reduction in fold height, the proliferation of crypt cells and the widening of mucosal lamina propria [24,25]. The shrinkage of the intestinal absorption area and the weakening of absorption capacity could ultimately lead to the impairment of growth and the reduction in feed convention [26]. In addition, the intestinal tract is the important immune organ and the most fragile immune barrier in fish [27]. However, glycinin could increase intestinal mucosal permeability, which might impair the immune system and augment the probability of antigens to invade their body, which then impacts other organs [16]. Lysozyme (LZM) and phosphatase secreted by immune cells play a vital part in the innate immunity of fish intestines [28]. It has been found that a high level of glycinin in feed would reduce LZM and phosphatase activities, leading to immune dysfunction [29]. Additionally, glycinin has also been reported to stimulate lipid peroxidation and protein oxidation. Studies on Jian carp verified that the intestine antagonized the oxidative stress induced by glycinin through increasing the transcriptional levels of antioxidant enzymes such as MnSOD, CuZnSOD, GPx1b and GPx4a. Nevertheless, these enzymes were easily inactivated by ROS, and, thus, the activities of antioxidant enzymes descended [16]. As a sensitizing protein, glycinin causes anaphylaxis and oxidative stress in juvenile aquatic animals, which negatively affects the development and health of the body. Moreover, oxidative stress surpassed the tolerance limit of the body, which will aggravate cell apoptosis [30]. Anti-apoptotic gene B-cell lymphoma 2 (Bcl-2) and pro-apoptotic gene Bcl-2-associated X (Bax) are the two most important genes in regulating fish cell apoptosis [31]. The release of mitochondrial Cyt C can activate Caspase 9 and caspase 3, which further mediates protein degradation and induces mitochondrial apoptosis [32]. Furthermore, the tight junction between intestinal epithelial cells is a vital structural basis to ensure the integrity of intestinal barrier function [33]. The disturbance of tight junctions in fish intestines may be reflected in the lower-regulated mRNA expression of barrier-forming tight junction proteins such as Cludinin-1, Occludin, and ZO-1 [34]. Interestingly, accumulating evidence has demonstrated that the damage of glycinin to the posterior intestine of aquatic animals is more severe than damage to the proximal and mid intestine [16,18,21,29]. Meanwhile, the uptake of macromolecular antigens by aquatic animals mainly occurs in the posterior intestine, accounting for 20–25% of the intestinal tract [35]. Therefore, posterior intestine is presumed to be the main site where glycinin acts.

Yellow catfish (*Pelteobagrus fulvidraco*) is a high-quality and valuable omnivorous economic fish, with tender meat and rich nutrients. Among them, hybrid yellow catfish (*Pelteobagrus fulvidraco* ♀ × *Pelteobaggrus vachelli* ♂) was obtained by artificial hybridization, with three consecutive generations of selected *Pelteobagrus fulvidraco* as female parent and two consecutive generations of selected *Pelteobaggrus vachelli* as male parent and showed better heterosis such as significantly improved growth rate and survival rate compared with common *Pelteobagrus fulvidraco* [36]. Recently, with the increase in hybrid yellow catfish culture, in order to reduce the feed cost, feed manufacturers have started to add large amounts of soybean meal to replace fish meal, causing the elevation of glycinin content in feed. Furthermore, the vigorous feeding of this species makes it easy to induce intestinal diseases marked by solitary swimming, abdominal enlargement, anal redness and swelling, and yellow mucus outflowed from the anus when gently pressing the abdomen. We hypothesized that the above aforementioned characteristics may be caused by the sensitizing damage of glycinin. Hence, the glycinin was isolated from soybean protein to explore its impacts on the growth and intestinal health of hybrid yellow catfish. Our results may provide a scientific theory foundation for promoting the efficiency of soybean protein utilization in aquaculture animals.

## 2. Results

### 2.1. Growth Performance and Feed Utilization

In this study, dietary glycinin levels considerably reduced FW and WGR of the fish (*p* < 0.05). Compared with the control group, 6% and 8% glycinin in diets significantly decreased the FI, SGR and PER (*p* < 0.05), while increased the FCR of the fish (*p* < 0.05). Obviously, the SR of fish in 4–8% glycinin groups was declined (*p* < 0.05). The increase in dietary glycinin linearly descended FW, FI, WGR, SGR, PER and SR (*p* < 0.05), and ascended the FCR of the fish (*p* < 0.05). Additionally, there was a quadratic response to PER of hybrid yellow catfish (*p* < 0.05). No obvious differences existed in HSI, VSI and CF (*p* > 0.05) (Table 1).

### 2.2. Nutritional Composition

Dietary glycinin levels linearly affected the moisture and crude lipid of whole-fish and muscle (*p* < 0.05) and caused a quadratic response to moisture and crude lipid of muscle (*p* < 0.05). Relative to the control group, 4–8% glycinin in diets obviously declined crude lipid (*p* < 0.05) and elevated the moisture of whole-fish (*p* < 0.05). Similarly, the crude lipid of muscle in the 8% glycinin group was considerably lower than that of other groups (*p* < 0.05), while the change in moisture content showed an opposite result. The crude protein and ash of whole-fish and muscle were not impacted by glycinin levels in diets (*p* > 0.05) (Table 2).

### 2.3. Morphology of Posterior Intestine and Liver

The morphological development indices in the posterior intestine of juvenile hybrid yellow catfish are shown in Table 3. With the rise in dietary glycinin levels, the relative height of fold depicted a descended trend (*p* < 0.05). Instead, the crypt depth in 4–8% glycinin groups was considerably higher than that of the control group (*p* < 0.05). Interestingly, the crypt depth in the 8% glycinin group was dramatically lower than that of the 4% and 6% glycinin groups (*p* < 0.05). The level of glycinin had significant linear and quadratic responses to the relative height of fold and the crypt depth of the posterior intestine (*p* < 0.05). No obvious differences were observed in the muscular thickness of the posterior intestine (*p* > 0.05).

The damage of glycinin on the posterior intestinal morphology of juvenile hybrid yellow catfish is shown in Figure 1. In the control group (a) and 2% glycinin group (b), the integrity of the intestinal structure was characterized by tight intestinal mucosal fold, arranged at alternate columnar epithelial cells and goblet cells, as well as uniform cells size, whereas the fish fed with 4% glycinin presented with posterior intestinal folds that were swollen and enlarged toward the lumen. Part of the epithelial cells were proliferated and disassociated. Furthermore, the goblet cells increased and the mucosal lamina propria became wider (c). Moderate lesions were found in the fish of 6% glycinin group including broken and shortened intestinal folds, proliferated and exfoliated epithelial cells and loose and severely separated lamina propria of the mucosa (d). Moreover, the fish fed dietary 8% glycinin displayed typical aspects of intestinal injury, such as seriously disrupted mucosal folds, acutely disordered columnar epithelial cells, significantly increased goblet cells, severely infiltrated lymphocytes, loose and broken lamina propria tissue and seriously separated intestinal mucosa (e).

The ultrastructure of posterior intestinal pathology of juvenile hybrid yellow catfish fed with different glycinin levels is presented in Figure 2. The mucosa fold is composed of epithelium, basement membrane and lamina propria. In the control and 2% glycinin groups, intestinal villi were neatly arranged. Intact organelles such as autolysosomes, mitochondria, and peroxisomes were expected and tight junctions also had intact and clearly identifiable structures (a,b). However, the striated margin of the intestinal villi and the structure of enterocyte membrane and mitochondria were destroyed in the 4% glycinin group (c). Intestinal villi in the 6% glycinin group was loose and disordered. The intestinal tight junction was partially fractured, and the cytoplasm became transparent (d). Indistinct tight junction, disorganized intestinal villi and severe cell vacuolation were found in the 8% glycinin group (e).

Microstructural observations showed that livers from juvenile hybrid yellow catfish fed with high-level glycinin were obviously damaged (Figure 3). The liver cell had a uniform size shape, with clear cell boundaries and central nuclei in the control group (a). However, liver cells of fish treated with 2% and 4% glycinin exhibited lipid vacuoles and a disorganized nucleus (b,c). Dietary 6% and 8% glycinin were accompanied by typical features of steatosis and injury of liver cells, including nuclear dissolution, nuclear shrinkage and nuclear offset. The number of lipid vacuoles also increased and the number of liver cells decreased compared with the control group (d,e).

### 2.4. Intestinal Nonspecific Immune and Antioxidant Enzyme Activity

The antioxidant and nonspecific immune enzyme activity in the posterior intestine of juvenile hybrid yellow catfish fed with different glycinin levels is presented in Figure 4. Glycinin levels in diets linearly effected the activities of LZM and T-SOD, and linearly and quadratically dropped the activities of ACP, AKP and the T-AOC (*p* < 0.05). Meanwhile, the LZM activity significantly descended in the 4–8% glycinin groups (*p* < 0.05), and the ACP activity presented a similar change in 8% glycinin groups (*p* < 0.05). In addition, dietary glycinin dramatically declined the AKP activity (*p* < 0.05). The T-AOC and the activity of T-SOD considerably decreased when diets contained 4–8% glycinin (*p* < 0.05).

### 2.5. Protein metabolic Enzyme Activity of Liver

Protein metabolism enzyme activities in the liver of juvenile hybrid yellow catfish are shown in Figure 5. ALT activity linearly correlated with dietary glycinin levels and was considerably lower in 4–8% glycinin groups than that of the control and 2% glycinin groups (*p* < 0.05). No obviously linear and quadratic differences in AST activity was found (*p* > 0.05). Additionally, diets with 6% and 8% glycinin induced significantly higher NO contents, and the content of NO showed linear and quadratic responses to glycinin levels of diets (*p* < 0.05).

### 2.6. Tight Junction mRNA Expression Levels of Posterior Intestine

The mRNA expression level of tight junction proteins in posterior intestine of juvenile hybrid yellow catfish fed diets with different glycinin levels are described in Figure 6. The mRNA levels of tight junction protein were linearly and quadratically effected by dietary glycinin levels (*p* < 0.05). Claudin-1 mRNA levels in the 6% and 8% glycinin groups and Occludin mRNA levels in the 4–8% glycinin groups were considerably declined (*p* < 0.05). Obviously, ZO-1 mRNA-level was downregulated by dietary glycinin (*p* < 0.05).

### 2.7. Apoptosis mRNA Expression Levels in Posterior Intestine

The mRNA expression levels of apoptosis-associated genes in the posterior intestine of juvenile hybrid yellow catfish are depicted in Figure 7. Bcl-2 mRNA expressions were quadratically impacted by dietary glycinin (*p* < 0.05) and remarkably downregulated by 8% dietary glycinin compared with the control group (*p* < 0.05). Glycinin levels in diets linearly affected Bax, Cyt C and p53 genes expressions (*p* < 0.05), and their levels considerably ascended in the groups of 6% and 8% glycinin (*p* < 0.05). In addition, diets with 6% and 8% glycinin induced dramatically higher caspase 3 and caspase 9 mRNA expressions, and their expression levels presented linear and quadratic responses to dietary glycinin levels (*p* < 0.05).

## 3. Discussion

Glycinin was recognized as one of the important reasons that hinder the extensive and effective use of soybean protein sources due to its high content and strong immunogenic in soybean. In this research, glycinin levels obviously affected the growth of juvenile hybrid yellow catfish, and a significant reduction in feed efficiency was observed when the dietary glycinin level was 6% and 8%. Similar results were reported that dietary 40–80 g/kg glycinin could suppress the growth of juvenile Amur minnow (*Rhynchocypris. lagowskii* Dybowski) [21], and the highest glycinin level could hinder feed utilization. Furthermore, dietary glycinin above 80 g/kg considerably decreased the survival rate and WGR of aquatic animals [16,19]. The presumable reason is that the tight molecular structure of glycinin in the natural state makes it difficult to be hydrolyzed by digestive enzymes. Macromolecular glycinin could not be decomposed, causing allergic reactions in fish intestine, which reduced the absorption efficiency of nutrients [19]. In particular, the hypoplastic digestive system of juvenile fish is fragile and it has a sensitive intestine, therefore it is more vulnerable to damage by glycinin. Moreover, glycinin may drop the palatability of feed, resulting in a growth reduction in hybrid yellow catfish. In the present study, dietary 6% and 8% glycinin considerably reduced FI of hybrid yellow catfish related to the control group. These results were also suggested in juvenile Jian carp [16], juvenile grass carp [20] and juvenile Amur minnow [21]. A study indicated that soybean antigen protein might change the expression level of T1R1 mRNA to affect the intestinal health, and finally affected FI of experimental subjects [37]. However, a different result was discovered that the growth of turbot is not harmed by dietary 8.31% glycinin [17]. The discrepancy might be due to the difference in the development degree of the digestive system. Although both of them were stomach fish, juvenile hybrid yellow catfish with an underdeveloped digestive tract have weaker degradability and tolerance to glycinin compared with turbot (initial weight, 9.98 g). More interestingly, the SR of hybrid yellow catfish fed glycinin diets was reduced dramatically related to the control group and consistent results have been observed in juvenile Chinese mitten crab [19]. While glycinin levels in the diet had no obvious influence on the SR of Jian carp [16], turbot [17] and Amur minnow [21]. Those distinct outcomes were partly contributed to the discrepancy in experimental conditions, subject specification and species.

In this study, dietary 4–8% glycinin descended the crude lipid and ascended the moisture of whole-fish. Additionally, 8% glycinin considerably decreased crude lipid and increased the moisture content of muscle. Similar effects of the high glycinin were presented in turbot [17]. Additionally, juvenile golden crucian carp fed with soy protein concentrate instead of fish meal had a consistent result [5]. The changes in nutritional composition indicated that high-level glycinin impacted the fat metabolism and reduced the fat deposition of juvenile fish. This might be associated with the cholesterol-lowering effect of glycinin. Previous studies have shown that the total and low-density lipoprotein cholesterol level in highly and mildly hypercholesterolaemic subjects could be cut by soybean plant protein food [38].

It is known that the integrity of the intestinal structure and function is an important guarantee for the normal development and healthy growth of aquatic animals. It is crucial to assess the potential effect of glycinin in feed on intestinal morphology. Intestinal barrier damage and dysfunction caused by glycinin had been proved in Jian carp [16], Chinese mitten crab [19], grass carp [20], Amur minnow [21] and obscure puffer [37]. Consistent results were found in our study that obviously shortened intestinal folds, increased crypt depth and destroyed intestinal structure were observed in 4–8% glycinin groups. This might be due to the indigestible glycinin that can easily pass through the intestinal mucosal barrier, then are captured by immune cells in the form of immunologic activity and produce antibodies. The immunoreactive substances stimulated a series of immune responses in fish, causing damage to the intestinal cell metabolism [17]. It well explained that dietary glycinin decreased the intestinal absorption area and reduced the effective utilization of nutrients, ultimately leading to the inhibition of fish growth performance. However, the crypt depth of the posterior intestine in the 8% glycinin group decreased to a certain extent compared with 4% and 6% glycinin groups. It was guessed that high-level glycinin damaged intestinal epithelial cells and inhibited cell proliferation. Further analysis on ultrastructure demonstrated that 4% to 8% glycinin in diets led to the unconsolidated, disordered and broken intestinal villi in varied degrees. The impaired intestine villi had a negative effect on the functions of intestinal digestion and absorption. Furthermore, mitochondrion is regarded as the central hub for the regulation of innate immunity and inflammation, and damaged mitochondrion may lead to immune imbalance and cell inflammation [39]. Additionally, transparent cytoplasm and destroyed organelles could result in the disorder of energy metabolism and the inhibition of protein synthesis [40].

To further explore the impact of dietary glycinin on the intestinal barrier and its mechanism, we detected the mRNA expression level of intestinal tight junction protein. Intestinal tight junction seals the gap between intestinal epithelial cells, which is the key to maintain the integrity of the intestinal physical barrier. Impaired intestinal tight junction could enhance epithelial permeability and increase pathogen sensitivity [41]. Transmembrane proteins claudin, occludin and plaque protein zonula (ZO) are important elements of intestinal tight junction [42]. Claudin-1 reduces paracellular permeability, and Occludin regulates the diffusion of small molecules in the intermembrane and pericellular, and ZO-1 binds directly to F-actin and participates in the formation of a cytoskeleton [41]. In the current study, the expression of the tight junction protein was affected by dietary glycinin. Among them, dietary 6% and 8% glycinin considerably inhibited the expression of Claudin-1, Occludin, and ZO-1 mRNA in the posterior intestine of hybrid yellow catfish. The result demonstrated that high-level glycinin seriously disrupted some of the tight junction components. This is consistent with the result that the destruction of the intestinal tight junction was observed in the ultrastructure. In addition, a few studies reported that glycinin exposure acutely impaired the intestinal tight junction in Jian carp [16] and grass carp [23], and the posterior intestine was more seriously impacted. This might be due to the fact that glycinin, as a foreign antigen, caused immune injury and inflammation, and then inflammatory factors destroyed intestinal epithelial tissue proteins. A great number of studies have proved that the destruction of the intestinal barrier function is directly related to inflammatory factors [16,43,44,45].

Earlier research has suggested that glycinin could damage the immunity and antioxidant systems of aquatic animals [17,18,21], and it is a primary cause of intestinal oxidative damage to the middle and posterior intestine [21]. Information on the immune function and oxidative stress can be reflected by the activities of non-specific immune and antioxidant enzymes, respectively [18,46]. LZM effectively prevents the release of endotoxin caused by bacteriolysis and inhibits the production of inflammatory factors by macrophages [47,48]. Phosphatases are involved in metabolic regulation and detoxification, in which ACP marks the existence of lysosomes, and AKP is the symbolic regulatory enzyme of phagocytes [21,29]. The ability of anti-oxidative stress could be examined by T-AOC [5]. SOD participates in the primary antioxidant protection of free radicals by converting superoxide anions into H_2_O_2_ and water to protect cells from potential ROS damage [49]. Additionally, the expression of antioxidant enzymes was positively correlated with the lifespan and health status of aquatic animals [6]. This study observed that the activities of LZM, T-AOC and the T-SOD were weakened in 4–8% glycinin groups. Compared to the control group, dietary 8% glycinin remarkably decreased the activities of ACP and AKP. The similar findings were found in the studies on different fish such as Amur minnow [21], golden crucian carp [18] and Jian carp [16]. It was demonstrated that the high-level dietary glycinin could inhibit the non-lifting immune and the anti-oxidative capacity of the aquatic animals. Similar results were obtained when fish meal was substituted with high-level soybean protein source in golden crucian carp [5] and abalone [6]. The extensive glycinin in feed used high proportion of soybean meal was one of the reasons for this consequence [18]. These negative affects might be caused by the fact that glycinin attacked fish intestine with strong immunogenicity, thus induced immune dysfunction and antioxidant damage. Study has confirmed that the high-level antigen protein would induce hepatopancreas impairment of Pacific white shrimp, and changed the susceptibility with organism to pathogen, so as to regulate the body’s immunity [29]. Furthermore, glycinin wreaked intestinal tissue and caused intestinal inflammation, which produced surplus ROS and resulted in oxidative damage [18]. When attacked by glycinin in the diet, the fish intestine synthesized more antioxidant enzymes from scratch to remove excessive ROS, but the antioxidant enzyme was constantly inactivated due to the instability [16,50]. Meanwhile, the existence of high-level glycinin in the diet could exacerbate in lipid peroxidation and protein oxidation, resulting in antioxidant capacity disturbance.

Results from the expression of apoptosis-linked genes demonstrated that 6% and 8% of glycinin enhanced apoptosis. The Bcl-2 protein family may affect the barrier function of the mitochondria membrane by forming pores [51], while pro-apoptotic proteins such as Bax participate in the formation or initiation of Cyt C channels through the localization and oligomerization in mitochondria [52]. When the dietary glycinin level reached 6%, the expression of Bcl-2 dropped, the expression of Bax ascended, and Cyt C was released, resulting in the initiation of caspase cascade reaction and the increase in caspase 3 and caspase 9 mRNA levels. A study on grass carp also indicated that glycinin might promote apoptosis, which exacerbates the damage of intestinal structure [23]. Additionally, glycinin elevated the apoptosis index of duodenum of piglets [53]. Additionally, the fish p53 gene affected both extrinsic and intrinsic apoptotic pathways [54], and p53 expression levels may elevate with phosphorylation under stress conditions [55]. P53 expression levels elevated linearly with rising dietary glycinin levels, indicating that glycinin could produce a certain promoting effect on apoptosis signaling. 

ALT and AST are critical transaminase that reflect the homeostasis of amino acid metabolism in fish, and they are also indicators of liver injury [56]. Our study observed that dietary 4–8% glycinin significantly decreased the activity of ALT in the liver. This revealed that the liver of juvenile hybrid yellow catfish was impaired to a certain extent, causing the depression of protein metabolism, which would affect the activity of protein metabolic enzymes [5,57]. Remarkably, when the body is under immunological stimulation (such as endotoxin, cytotoxin), large amounts of NO released by macrophages react with superoxide anion and other free radicals, resulting in cytotoxic effect, inducing or enhancing tissue pathological changes [58,59]. The content of NO in liver was considerably increased by dietary 6% and 8% glycinin in this study, which is probably because indigestible glycinin-induced inflammatory factors stimulated macrophages to release excessive NO, causing damage to cells and tissues of the liver. A report confirmed that the level of NO increased significantly in acute liver damage caused by non-steroidal anti-inflammatory drugs, and the change trend of NO level was the opposite, with the activity of SOD, indicating that NO may induce liver damage by damaging the antioxidant system [60]. It was speculated that the infiltration of glycinin could cause the disorder of liver immunity and the antioxidant system, while further research should be needed to confirm that the relationship and mechanism between the changes in NO and the activity of antioxidant enzymes caused by glycinin. In order to further verify liver damage, our study observed histomorphological sections. The results showed that 6% and 8% of glycinin in diets induced liver lesions, including the significant increase in lipid vacuoles. Additionally, consistent results have been reported in Amur minnow [21]. Likewise, lipid accumulation and inflammation were observed in the liver of totoaba when the dietary soybean meal was included above 22% [61], and the liver of juvenile starry flounder was damaged when more than 60% soy protein concentrate was ingested [62]. Liver is a special and vital organ in the process of material and energy metabolism. The liver fat of fish majorly comes from the direct absorption of feed fatty, as well as the transformation of excess protein and carbohydrates in feed [63]. If liver fats were not transported or converted into lipoproteins, it would accumulate lipids and cause metabolic disorders of the liver [21,64]. Furthermore, the experimental results suggest that the influence of high-level glycinin on crude lipid contents of whole-body and muscle in the present study were also consistent with liver injury. Of course, there are a few reports on the specific mechanism of glycinin-induced liver injury, and more in-depth research is still needed.

As is aforementioned, dietary glycinin could inhibit the growth of juvenile hybrid yellow catfish fish (initial weight, 1.02 ± 0.01 g). When the level of dietary glycinin reached 6%, feed utilization was significantly reduced. Moreover, glycinin level in the diet over 4% would impair the structures of the posterior intestine and liver and weaken posterior intestine immune ability and antioxidant capacity. A higher level of glycinin in diets (> 6%) would contribute to apoptosis in the posterior intestine of juvenile hybrid yellow catfish.

## 4. Materials and Methods

### 4.1. Glycinin and Diet Preparation

Glycinin was separated from commercial soybeans (Wuhan Alpha Agri-tech Co., Ltd., Wuhan, China) using the isoelectric point precipitation method described by Thanh and Shibasaki [65], but with specific modifications. In brief, soybean protein was extracted from soybean powder defatted by n-hexane using 0.02 mol/L calcium dihydrogen phosphate. NaCl was added to the protein separation solution and its final concentration was 50 mmol/L (Cat. No. 40066289, 73522260, 20012418, Sinopharm Chemical Reagent Co., Ltd., Shanghai, China). The final solution was adjusted pH and centrifuged. The supernatant was precipitated at 4 °C overnight to separate glycinin. The purity of glycinin measured by plant glycinin enzyme-linked immunoassay kit (Jiangsu Jingmei Biological Technology Co., Ltd., Yancheng, China) was 85.62%. Its water content was determined and dried at 105 °C until a constant weight and was 72.59%. Figure 8. presents the sodium dodecyl sulphate and polyacrylamide gel (SDS-PAGE) electrophoresis diagram of the isolated glycinin.

Five isonitrogeous (42% crude protein) and isolipidic (9% crude fat) diets were made to meet the nutritional requirements of the experimental fish (Table 4). Fish meal, chicken powder and corn gluten flour were chosen as basic feed protein resources. The level of glycinin in the control group was 0%. Another four diets were added 2%, 4%, 6% and 8% glycinin, respectively, which was equivalent to 20%, 40%, 60% and 80% of fish meal proteins substituted by soybean meal proteins, and other soybean meal protein components were balanced with casein. Soybean phospholipid oil was applied as the main fat source. Feed raw materials were provided by Wuhan Alpha Agri-tech Co., Ltd., (Wuhan, China). Before the process of marking experimental diets, these materials were sieved using a 425 μm sifter. Small materials (such as adhesives and antioxidants) were mixed step by step, and then blended into large materials (such as protein raw materials and flour). Subsequently, phospholipid oil was added. The above mixtures were stirred with a mixer for 30 min and then 300 g of distilled water was added per kilogram of ingredients. A screw extruder (Hobart Mixer, Model A200) was utilized to squeeze the dough into particles with a 1.0 mm diameter. They were air-dried in an oven at 45 °C until the moisture content was lower than 10%. All diets were put into a self-sealed bag and placed in a −20 °C refrigerator until used and tested. The immunological activity of glycinin was detected with a plant glycinin ELISA kit provided by Jingmei Biological Technology Co., Ltd. (Yancheng, China). They were 0%, 1.74%, 3.57%, 5.45%, and 7.27% in the groups of 0%, 2%, 4%, 6% and 8%, respectively.

### 4.2. Growth Trial and Rearing Facilities

Hybrid yellow catfish were provided by Hubei Huangyouyuan Fisheries Development Co. Ltd., Wuhan, China. The breeding experiment was performed at the circulating water culture system in the aquaculture greenhouse of Huazhong Agricultural University aquaculture base (Wuhan, China). Five hundred uniform-sized and healthy hybrid yellow catfish were selected and acclimated in fiberglass tanks (1.20 × 0.60 × 0.45 m, the water depth was kept at 0.30~0.35 m) for two weeks. Subsequently, 450 fish (1.02 ± 0.01 g initial body weight) were randomly allotted to 5 groups, 30 fish per tank and 3 tanks per group. Each group experimental diet was randomly fed into triplicate tanks, and the experimental fish were fed twice a day at 08:00 and 18:00, with a feed ration of 2.0~2.5% of their body weight per meal. Three hours after feeding each meal, uneaten feed was collected with a siphon and dried to calculate feed intake (FI). Fresh water of 1/3 was changed every two days to ensure good water quality. The water quality was monitored daily during the experiment and the breeding water temperature was kept from 24.0 °C to 28.0 °C, dissolved oxygen was >5.2 mg L^−1^, total ammonia nitrogen was <0.28 mg L^−1^ and pH was kept at 7.4 ± 0.2.

### 4.3. Sample Collection

After eight weeks of the feeding experiment, all fish were fasted for 24 h and anesthetized with MS-222 solution. Final weight, survival number and body length were recorded to estimate the corresponding indices of growth and feed utilization. Five fish were randomly selected from every trial tank and placed in a −20 °C refrigerator for determining the routine nutritional indications of whole fish. In addition, 6 fish were taken from each tank to separate viscera and liver under aseptic operation and weighed to compute the physique indices. Liver, muscle and posterior intestine tissue from 9 fish were chosen at random in each tank. All samples were frozen by liquid nitrogen and refrigerated to −80 °C to detect nutrients, activities of antioxidant and immune enzymes as well as gene expression. Posterior intestines and liver from six fish in each experimental group were dissected and fixed in neutral paraformaldehyde solution at room temperature for 24 h, then sections were made to evaluate the damage of the morphological structure. Posterior intestines of one fish in each experimental group were fixed in 2.5% glutaraldehyde for transmission electron microscopic observation of the intestinal ultrastructure.

### 4.4. Calculation Formula of Growth Indexes

The parameters were computed as the following:Feeding intake (FI, %/d) = 100 × feed consumption × 2/[(W_f_ + W_i_) × d] 
Weight gain rate (WGR, %) = 100 × [W_f_ – W_i_]/W_i_

Specific growth rate (SGR, %/day) = 100 × [(Ln W_f_ – Ln W_i_)/d]
Feed coefficient rate (FCR) = Dry feed intake/(W_f_ − W_i_) 
Protein efficiency ratio (PER, %) = 100 × (W_f_ − W_i_)/(dietary protein intake) 
Survival rate (SR, %) =100 × (final fish number/initial fish number)
Hepatosomatic index (HSI, %) = 100 × weight of liver/W_f_
Viscerosomatic index (VSI, %) = 100 × weight of viscera/W_f_
Condition factor (CF, g /cm^3^) = 100 × W_f_ (g)/[body length of fish (cm)]^3^

W_f_: final weight of fish; W_i_: initial weight of fish; d: feeding days.

### 4.5. Nutritive Composition Analysis

The nutritional composition of muscle, whole fish and diet was analyzed by the following methods (AOAC, 1995): moisture was determined by drying at 105 °C until a constant weight; crude protein (N × 6.25) was assayed using an automatic Kjeldahl nitrogen analyzer (2300 Auto analyzer, Foss Tecator, AB, Hoganas, Högsby, Sweden); crude fat was analyzed by Soxhlet extraction; and crude ash was determined by high temperature combusting at 550 °C.

### 4.6. Histological Assessment

The histopathological sections of the posterior intestine and liver were conducted as Kissinger et al. description [66]. The fixed tissues were dehydrated with ethanol and dipped in paraffin after xylene equilibrium. Subsequently, samples were embedded, sliced, and then stained using hematoxylin-eosin. After the histo-sections were encapsulated by neutral gum, they were observed under optical microscope and measured and analyzed by MShot Image Analysis System 1.1.5 software. In order to clarify the damage degree of glycinin to posterior intestine, the height of mucosal fold related to intestinal diameter, crypt depth and thickness of the muscle layer in each repetitive group were measured. For ultrastructural observation, the posterior intestine fixed in 2.5% glutaraldehyde was dehydrated by a series of ethanol (30%, 50%, 70%, 80%, 95%, 100%), and then embedded with embedding agent in a 37 °C oven overnight. After polymerization at 60 °C for 48 h, they were sliced with an ultramicrotome (Leica UC7, Leica, Shanghai, China). The above procedures were completed by Servicebio Biotechnology Co., Ltd. (Wuhan, China). After dyeing and drying, the images of ultrathin slices were collected and analyzed under transmission electron microscope (HT7800, HITACHI, Dongjing, Japan).

### 4.7. Enzyme Activity Determination

The following tests were performed for the enzyme activity of intestine and liver. First, 0.1 g samples were mechanically homogenized in 10 volumes precooled 0.65% normal saline and were centrifugated at 4 °C with 2500× *g* (5415R, Eppendorf, Hamburg, Germany) for 15 min. The supernatant was collected and placed in −80 °C until detection. Intestinal enzyme activities including total antioxidant capacity (T-AOC), total superoxide dismutase (T-SOD), acid phosphatase (ACP), alkaline phosphatase (AKP), lysozyme (LZM) and the liver of glutamic oxaloacetic transaminase (AST), glutamic pyruvic transaminase (ALT) activities and nitric oxide (NO) content were tested through the kit of Nanjing Jiancheng Bioengineering Institute, Nanjing, China. Specific testing and calculation methods were carried out as instructed.

### 4.8. Analysis of Gene Expression in Posterior Intestine

Posterior intestine tissue (20 mg) was homogenized in RNA extract solution (Servicebio: G3013) using a tissue grinder (KZ-III-F, Servicebio, Wuhan, China) in a 2 mL centrifugal tube. The supernatant after centrifugation was extracted with chloroform and the RNA was precipitated with isopropanol. Finally, the product was washed with anhydrous ethanol (Cat. No. 10006818, 80109218, 10009218, Sinopharm Chemical Reagent Co., Ltd.). The concentration and purity of RNA were measured with a Nanodrop 2000 spectrophotometer (IMPLEN). cDNA was synthesized with the SweScript RT I First Strand cDNA Synthesis Kit (Servicebio, Wuhan, China) and placed at −80 °C for subsequent detection. All primers were designed in NCBI and synthesized by Shenggong Bioengineering Co., Ltd. (Shanghai, China) (Table 5). Quantitative real time-PCR (RT-qPCR) analysis was conducted using the QuantStudioTM 6 Flex System (Life technologies, Grand Island, USA) according to 2×SYBR Green qPCR Master Mix (None ROX) (Servicebio, Wuhan, China) instructions. Every sample was tested in triplication. Values were quantified using the 2^–ΔΔCT^ method [67] and a statistical analysis was carried out.

### 4.9. Statistical Analysis

Data were submitted to SPSS statistics 16.0 software. All data were verified for normality and homogeneity of variances by Shapiro–Wilk and Levene’s tests, respectively. Under the conditions of normal distribution and variance homogeneity test, differences between experimental groups with different levels of glycinin were analyzed by one-way analysis of variance (ANOVA). Final data were expressed by “means ± standard error” (means ± SE). The multiple range test analysis was determined by Tukey’s method. Additionally, linear and quadratic responses of glycinin levels to hybrid yellow catfish were detected by the orthogonal polynomial contrast method. Significant differences were *p*-value lower than 0.05.

## Figures and Tables

**Figure 1 ijms-23-11198-f001:**
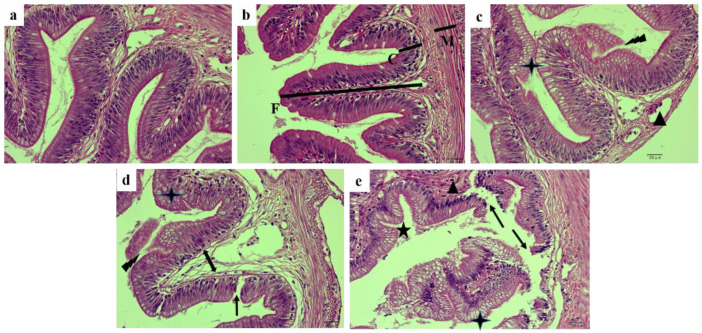
H&E-stained section of the posterior intestinal from hybrid yellow catfish fed with different glycinin levels. (**a**), the control; (**b**), 2% glycinin; (**c**), 4% glycinin; (**d**), 6% glycinin; (**e**), 8% glycinin. Triangles show lymphocyte infiltration; four-pointed stars show goblet cell hyperplasia; five-pointed stars show the bending of intestinal folds; lightning shapes show the proliferation and exfoliation of intestinal epithelial cells; single arrows show disrupted intestinal folds; double arrows show the separation of the inherent layer. F, Height of fold; C, Crypt depth; M, Thickness of muscle layer. Magnification was 400×, and the scale represents 20 μm.

**Figure 2 ijms-23-11198-f002:**
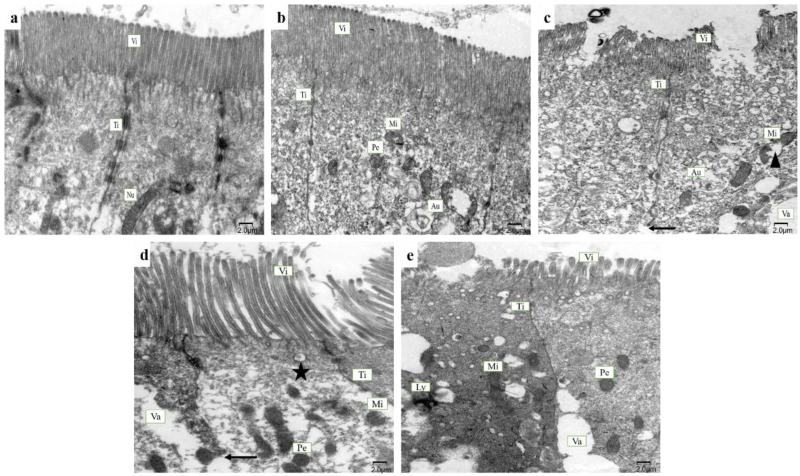
Ultrastructure of posterior intestinal cells of hybrid yellow catfish fed with different glycinin levels under electron microscope. Intestinal villi from fish of control group showing neat arrangement (**a**). Posterior intestine from fish of 2% glycinin group have clear tight junctions between enterocytes, with abundant organelles (**b**). However, posterior intestine from fish of 4% glycinin group showing mild microvesicular vacuolation and intestinal villous striated margin shedding, as well as mitochondria damage (triangle) and broken tight junction (arrow) (**c**). Enterocytes from fish of 6% glycinin group displaying moderate vacuolation, with cytoplasmic transparency, as well as loose intestinal villi and broken tight junction (arrow) (**d**). Intestinal villi from fish of 8% glycinin group disorganized and enterocytes appearing serious vacuolation (**e**). Five-pointed star indicating fusion morphology of secondary lysosome and phagocytic vesicles. Vi, intestinal villi; Ti, tight junction; Nu, nucleus; Mi, mitochondria; Pe, peroxisome; Au, autolysosome; Va, vacuolation; Ly, lysosome. The scale represents 2 μm.

**Figure 3 ijms-23-11198-f003:**
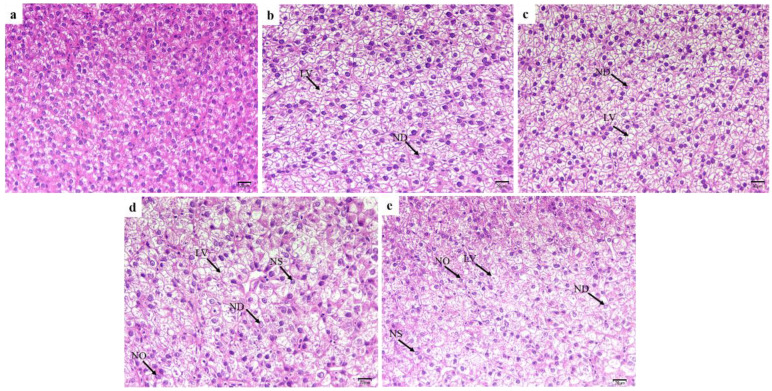
H&E-stained sections of liver from hybrid yellow catfish fed with different glycinin levels. (**a**), the control; (**b**), 2% glycinin; (**c**), 4% glycinin; (**d**), 6% glycinin; (**e**), 8% glycinin. LV: lipid vacuoles; ND: nuclear dissolution; NS: nuclear shrinkage; NO: nuclear offset. Magnification was 400×, and the scale represents 20 μm.

**Figure 4 ijms-23-11198-f004:**
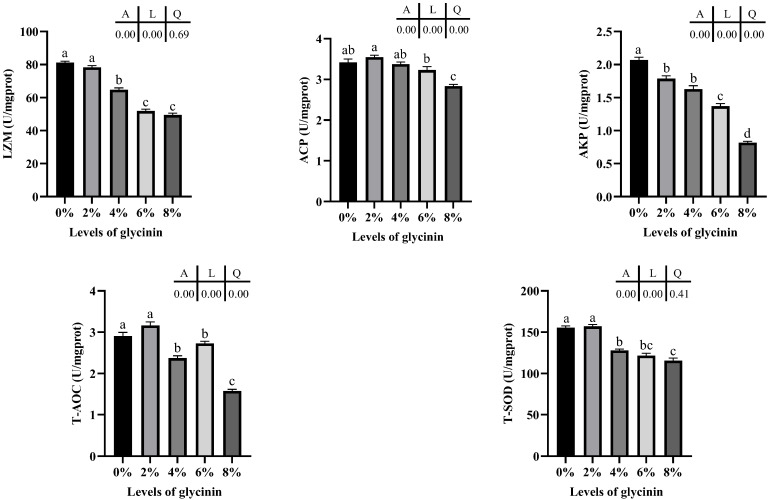
The antioxidant and nonspecific immune enzyme activity in posterior intestine of hybrid yellow catfish fed with different glycinin levels. Data describe mean ± SE (n = 3). Obvious difference of means between groups is indicated by a, b, c and d (*p* < 0.05). A represent the variance analyzed by one-way ANOVA; L represent linear trend analyzed by orthogonal polynomial contrasts; Q represent quadratic trend analyzed by orthogonal polynomial contrasts.

**Figure 5 ijms-23-11198-f005:**
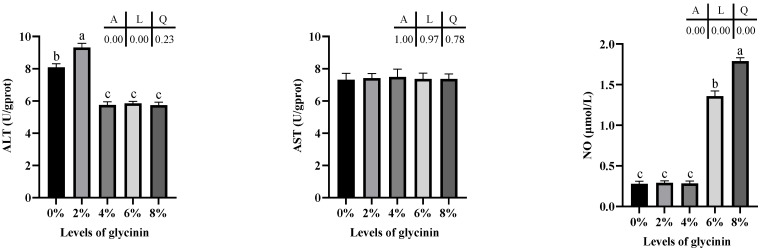
Protein metabolism enzyme activities in liver of hybrid yellow catfish fed with different glycinin levels. Data describe mean ± SE (n = 3). Obvious difference of means between groups is indicated by a, b and c (*p* < 0.05). A represent the variance analyzed by one-way ANOVA; L represent linear trend analyzed by orthogonal polynomial contrasts; Q represent quadratic trend analyzed by orthogonal polynomial contrasts.

**Figure 6 ijms-23-11198-f006:**
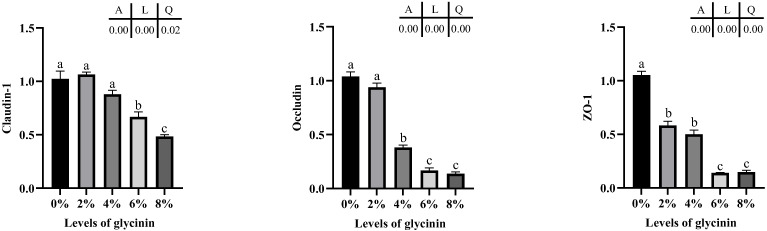
Relative mRNA expression of Claudin-1, Occludin and ZO-1 in posterior intestine of hybrid yellow catfish fed with different glycinin levels. Relative mRNA expression was evaluated by RT-qPCR. Data describe mean ± SE (n = 3). Obvious difference of means between groups is indicated by a, b and c (*p* < 0.05). A represent the variance analyzed by one-way ANOVA; L represent linear trend analyzed by orthogonal polynomial contrasts; Q represent quadratic trend analyzed by orthogonal polynomial contrasts.

**Figure 7 ijms-23-11198-f007:**
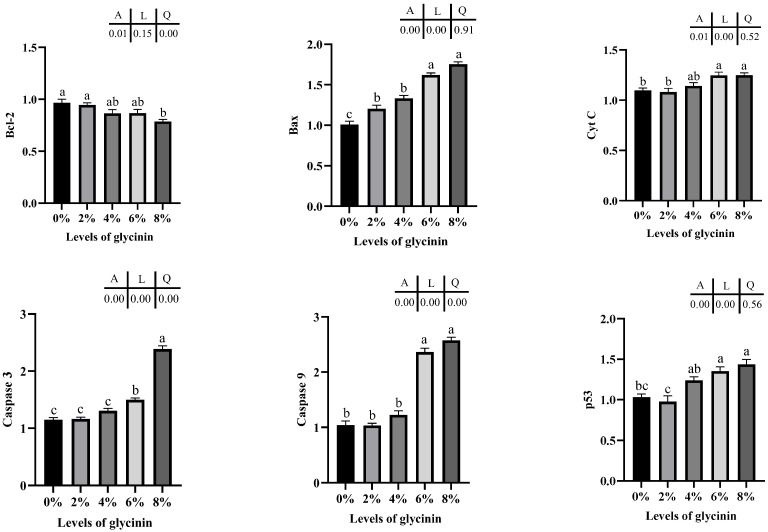
Expressions of apoptosis associated genes in posterior intestine of hybrid yellow catfish fed with different glycinin levels. Relative mRNA expression was evaluated by RT-qPCR. Data describe mean ± SE (n = 3). Obvious difference of means between groups is indicated by a, b and c (*p* < 0.05). A represent the variance analyzed by one-way ANOVA; L represent linear trend analyzed by orthogonal polynomial contrasts; Q represent quadratic trend analyzed by orthogonal polynomial contrasts.

**Figure 8 ijms-23-11198-f008:**
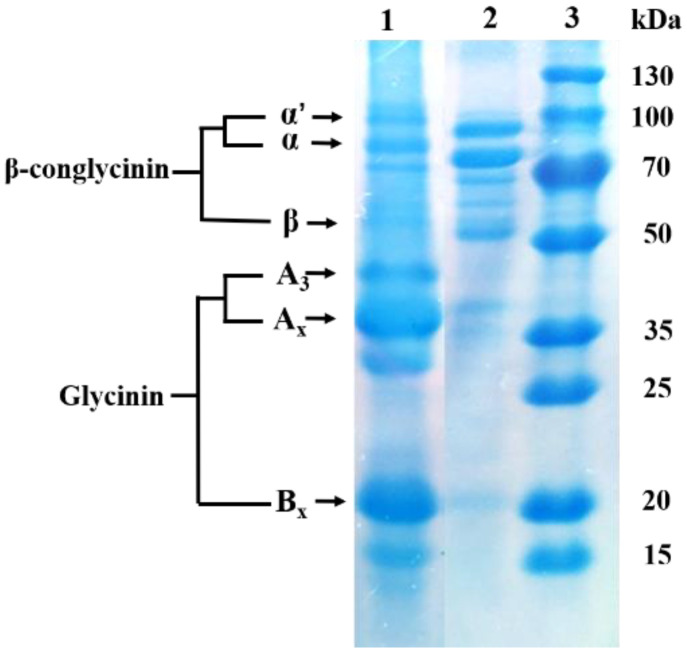
SDS-PAGE diagram of isolated glycinin. Lane 1: glycinin; Lane 2: β-conglycinin; Lane 3: protein marker. Ax and Bx represent acidic and basic subunits of isolated glycinin. α′, α, β stand for subunit fractions of β-conglycinin.

**Table 1 ijms-23-11198-t001:** Growth performance and feed utilization of trial ingredients for hybrid yellow catfish.

	Levels of Glycinin					Pr > F ^1^		
	0%	2%	4%	6%	8%	ANOVA	Linear	Quadratic
IW (g)	1.03 ± 0.02	1.02 ± 0.01	1.02 ± 0.01	1.04 ± 0.01	1.01 ± 0.01	0.26	0.85	0.72
FW (g)	6.82 ± 0.37 ^a^	5.85 ± 0.11 ^b^	5.67 ± 0.06 ^b^	3.76 ± 0.29 ^c^	3.34 ± 0.20 ^c^	0.00	0.00	0.49
FI (%/d)	2.90 ± 0.03 ^a^	2.80 ± 0.02 ^ab^	2.78 ± 0.02 ^ab^	2.77 ± 0.05 ^b^	2.70 ± 0.02 ^b^	0.01	0.00	0.50
WGR (%)	562.27 ± 25.75 ^a^	475.09 ± 15.95 ^b^	458.27 ± 5.25 ^b^	261.02 ± 29.72 ^c^	229.89 ± 17.16 ^c^	0.00	0.00	0.40
SGR (%/d)	3.20 ± 0.06 ^a^	2.96 ± 0.05 ^a^	2.91 ± 0.04 ^a^	2.16 ± 0.14 ^b^	2.02 ± 0.08 ^b^	0.00	0.00	0.14
FCR	1.20 ± 0.01 ^b^	1.17 ± 0.02 ^b^	1.27 ± 0.01 ^b^	1.65 ± 0.08 ^a^	1.83 ± 0.18 ^a^	0.00	0.00	0.06
PER (%)	1.90 ± 0.02 ^a^	1.95 ± 0.03 ^a^	1.88 ± 0.02 ^a^	1.47 ± 0.05 ^b^	1.35 ± 0.04 ^b^	0.00	0.00	0.00
SR (%)	87.22 ± 0.96 ^a^	87.77 ± 1.93 ^a^	76.67 ± 1.28 ^b^	78.33 ± 2.25 ^b^	71.44 ± 2.75 ^b^	0.00	0.00	0.76
VSI (%)	8.02 ± 0.85	8.17 ± 0.77	8.25 ± 0.83	7.99 ± 0.81	9.09 ± 0.80	0.86	0.46	0.62
HSI (%)	2.11 ± 0.19	2.08 ± 0.18	2.02 ± 0.12	1.99 ± 0.19	2.04 ± 0.23	0.99	0.70	0.79
CF (g/cm^3^)	1.66 ± 0.04	1.70 ± 0.08	1.65 ± 0.04	1.56 ± 0.02	1.70 ± 0.08	0.46	0.72	0.54

Data describe mean ± SE (n = 3). Obvious difference of means within same line is indicated by a, b and c (*p* < 0.05). ^1^ Significance probability associated with the F-statistic. IW: initial weight; FW: final weight; FI: feeding intake; WGR: weight gain rate; SGR: specific growth rate; FCR: feed coefficient rat; PER: protein efficiency ratio; SR: survival rate; VSI: viscerosomatic index; HSI: hepatosomatic index; CF: condition factor.

**Table 2 ijms-23-11198-t002:** Nutritional components in whole fish and muscle of hybrid yellow catfish (air-dry basis) (%).

	Levels of Glycinin					Pr > F ^1^		
	0%	2%	4%	6%	8%	ANOVA	Linear	Quadratic
Whole fish								
Moisture	75.09 ± 0.24 ^d^	75.44 ± 0.26 ^cd^	76.30 ± 0.24 ^bc^	77.12 ± 0.22 ^ab^	77.54 ± 0.24 ^a^	0.00	0.00	0.92
Ash	3.54 ± 0.03	3.36 ± 0.06	3.35 ± 0.05	3.43 ± 0.14	3.31 ± 0.15	0.55	0.25	0.58
Crude lipid	5.65 ± 0.12 ^a^	5.20 ± 0.12 ^a^	4.50 ± 0.16 ^b^	3.79 ± 0.08 ^c^	3.79 ± 0.0.09 ^c^	0.00	0.00	0.07
Crude protein	15.10 ± 0.16	14.85 ± 0.32	14.84 ± 0.09	14.57 ± 0.34	14.61 ± 0.33	0.64	0.16	0.76
Muscle								
Moisture	79.86 ± 0.22 ^ab^	79.36 ± 0.21 ^b^	79.42 ± 0.26 ^b^	79.59 ± 0.12 ^b^	80.63 ± 0.24 ^a^	0.01	0.03	0.00
Ash	1.27 ± 0.05	1.33 ± 0.02	1.29 ± 0.01	1.33 ± 0.01	1.25 ± 0.03	0.26	0.66	0.17
Crude lipid	1.94 ± 0.09 ^b^	2.47 ± 0.09 ^a^	2.10 ± 0.05 ^b^	1.89 ± 0.06^b^	1.20 ± 0.08 ^c^	0.00	0.00	0.00
Crude protein	16.33 ± 0.32	16.25 ± 0.18	16.48 ± 0.21	16.40 ± 0.29	15.97 ± 0.13	0.39	0.47	0.14

Data describe mean ± SE (n = 3). Obvious difference of means within same line is indicated by a, b, c and d (*p* < 0.05). ^1^ Significance probability associated with the F-statistic.

**Table 3 ijms-23-11198-t003:** Posterior intestinal morphology indexes of hybrid yellow catfish.

	Levels of Glycinin					Pr > F ^1^		
	0%	2%	4%	6%	8%	ANOVA	Linear	Quadratic
Relative height of fold (%)	66.23 ± 0.43 ^a^	59.11 ± 0.55 ^b^	50.80 ± 0.53 ^c^	50.68 ± 0.63 ^c^	48.47 ± 0.71 ^d^	0.00	0.00	0.00
Crypt depth (μm)	31.69 ± 0.46 ^c^	28.88 ± 0.46 ^d^	38.09 ± 0.50 ^a^	39.40 ± 0.61 ^a^	33.74 ± 0.44 ^b^	0.00	0.00	0.00
Thickness of muscle layer (μm)	27.78 ± 0.36	27.59 ± 0.49	26.85 ± 0.46	27.53 ± 0.42	27.45 ± 0.64	0.70	0.63	0.37

Data describe mean ± SE (n = 3). Each replicate value is means of two experimental fish from each replicate group (8 measurements per fish). Obvious difference of means within same line is indicated by a, b, c and d (*p* < 0.05).^1^ Significance probability associated with the F-statistic.

**Table 4 ijms-23-11198-t004:** Formulation and nutritional composition of experimental feeds (air-dry basis).

Ingredients (%)	Levels of Glycinin				
	0%	2%	4%	6%	8%
White fishmeal	40.00	32.00	24.00	16.00	8.00
Chicken powder	10.00	10.00	10.00	10.00	10.00
Corn gluten flour	8.00	8.00	8.00	8.00	8.00
Glycinin	0.00	2.08	4.16	6.24	8.32
Casein	0.00	2.72	5.44	8.16	10.88
Soybean phospholipid oil	5.00	5.60	6.20	6.80	7.40
High gluten flour	27.50	30.10	32.70	35.30	37.90
Choline chloride	1.00	1.00	1.00	1.00	1.00
Vitamin premix ^1^	2.00	2.00	2.00	2.00	2.00
Mineral premix ^2^	2.00	2.00	2.00	2.00	2.00
Sodium alginate	2.00	2.00	2.00	2.00	2.00
Antioxidant	0.50	0.50	0.50	0.50	0.50
Anti-mildew agent	0.50	0.50	0.50	0.50	0.50
Betaine	1.00	1.00	1.00	1.00	1.00
Chromium trioxide	0.50	0.50	0.50	0.50	0.50
Nutritional composition					
Moisture	12.59	12.96	12.51	12.92	12.88
Crude protein	42.41	42.19	42.39	42.17	42.62
Crude lipid	9.23	9.14	9.73	9.13	9.38
Ash	8.80	8.92	8.71	8.30	8.14

^1^ Vitamin premix (mg/kg diet): Vitamin A, 1.67; Vitamin D, 0.027; Vitamin E, 50.20; Vitamin K, 11.10; Vitamin C, 100.50; Folic acid, 5.20; Calcium patothenate, 50.20; Inositol, 100.50; Niacin, 100.50; Biotin, 0.12; Cellulose, 645.25. ^2^ Mineral premix (mg/kg diet): NaCl, 500.15; MgSO_4_·7H_2_O, 8155.55; NaH_2_PO_4_·2H_2_O, 12,500.51; KH_2_PO_4_, 16,000.51; Ca(H_2_PO_4_)_2_·H_2_O, 7650.50; FeSO_4_·7H_2_O, 2286.15; C_6_H_10_CaO_6_·5H_2_O, 1750.12; ZnSO_4_·7H_2_O, 178.12; MnSO_4_·H_2_O, 61.35; CuSO_4_·5H_2_O, 15.45; CoSO_4_·7H_2_O, 0.89; KI, 1.5; Na_2_SeO_3_, 0.59.

**Table 5 ijms-23-11198-t005:** Primer used in this study.

Gene Name		Primer Sequence (5′ to 3′)	GenBank Number
Claudin-1	Forward	ACGCTAACAACGGCTCAGA	XM_027148337.1
	Reverse	CCTTACATTCAGACACCACCTT	
Occludin	Forward	CGAGCGAGAGACTACGACAC	XM_027141818.1
	Reverse	TCCAGGAATTGTGGGCTTCC	
ZO-1	Forward	GCGAACTCTCTGAACAGCCT	XM_027154070.1
	Reverse	TGTGTGTGTGCAGGAGGTTT	
Bcl-2	Forward	TTTAGACAAACGGGGCAGGG	XM_027170693.1
	Reverse	AGCTCCATCATCTGCCCCTA	
Bax	Forward	TTTCTGGCCGGAGTTCTCAC	XM_027161178.2
	Reverse	TGGATCTGCCCCTTATTGCC	
Cyt C	Forward	GCAGGATACGAGCAAGAT	XM_027159893.2
	Reverse	TACACGGATGCCAACAAG	
Casepase 3	Forward	GTCGTGAGGCAGTCGGTATT	XM_027134813.2
	Reverse	TAATGGCTCGGATGCAGGTC	
Casepase 9	Forward	GCTTCCCTGGAGCAGTTCAT	XM_047807446.1
	Reverse	TGCTGTACCTCATCGGGAGA	
p53	Forward	CTTCCTACAGGCTTTAGACAA	XM_047808967.1
	Reverse	GTAAGAAATCCAAGAACACCA	
β-actin	Forward	CCACTTCCCCCACATCACTG	XM_027142941.1
	Reverse	AGAACCCCAGGAAAGCTCACG	

## Data Availability

Not applicable.

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
