# Peer review of "Effects of Dietary Glycinin on Oxidative Damage, Apoptosis and Tight Junction in the Intestine of Juvenile Hybrid Yellow Catfish, Pelteobagrus fulvidraco ♀ × Pelteobaggrus vachelli ♂"

_ijms, 2022, doi:10.3390/ijms231911198_

Round 1
Reviewer 1 Report
The manuscript ijms-1917966 entitled “Glycinin disturbed intestinal structural integrity related to oxidative damage, aggravation of apoptosis and downregulated transcription of tight junction in the intestine of juvenile hybrid yellow catfish, Pelteobagrus fulvidraco ♀ × Pelteobaggrus vachel” provides comprehensive scientific results about the influences of glycinin on the growth and intestinal structural integrity related to oxidative damage, apoptosis and tight junction of juvenile hybrid yellow catfish. In general, the topic is important for the aquaculture industry. The manuscript was well written and well structured. However, there are several errors in the English language and style. I just highlighted some of them. The manuscript must undergo revision by a language editing service or a native speaker.
I would like to recommend a minor revision and my specific comments are as below:
Abstract:
Line 17-18: removed “, respectively”
Line 21-22: “which significant reductions showed in 4%-8% glycinin groups”. This sentence is not clear to me and must be revised.
Results
Line 300-301: “No obvious differences in muscular thickness of posterior intestine (P > 0.05).” there is no verb in this sentence.
Line 343: Fig 3 is not clear the resolution is low. Authors must provide high-quality Fig
Line 329: There is no scale in this figure
Line 354: “The scale represents 2μm”. There is no scale in this figure
Line 358: “and the scale represents 20μm.” I can't see the scale in the figure. The scale must be presented in the figure
Discussion
Line 435: change “effected” to “affected”
Line 473: “As far as all know” This sentence is very suitable for scientific writing. Please revise it.
Line 477: “by” must be replaced by “in”
Reviewer 2 Report
Title needs to be short and straight. Please revise
Line 17: % (w/w or w/v?)
Line 19: small letter for p < 0.05. Change throughout the manuscript.
Line 21: Does not make clear sense. Rewrite the sentence.
Does the Abstract length meet journal guideline? Check word limit.
Use different keywords rather than title.
Line 38: Soybean proteins are considered
Line 20: Add a section "Chemicals" and add chemical details including the company, purity...
Line 122: Add reference for the method
line 135: wet weight or dry weight basis?
Line 138: % (w/w, w/v?)
Line 169: After the experimental period, fish were
Line 171: When did you remove the unused feed? daily?
Line 172: Did you measure water quality parameters daily or weekly?
Table 3: Mention all the abbreviations in footnote.
